# Antimicrobial Activity and Mechanism of Action of the *Amaranthus tricolor* Crude Extract against *Staphylococcus aureus* and Potential Application in Cooked Meat

**DOI:** 10.3390/foods9030359

**Published:** 2020-03-19

**Authors:** Ling Guo, Yanyan Wang, Xue Bi, Kai Duo, Qi Sun, Xueqi Yun, Yibo Zhang, Peng Fei, Jianchun Han

**Affiliations:** 1Key Lab of Dairy Science, Ministry of Education, College of Food Science, Northeast Agricultural University, Harbin 150030, China; guoling@neau.edu.cn (L.G.); wyy1792830896@163.com (Y.W.); xuejingupup@163.com (X.B.); 15754505440@163.com (Q.S.); vaeyxq009@163.com (X.Y.); Yibo__Zhang@163.com (Y.Z.); 2Heilongjiang Institute for Food and Drug Control, Harbin 150001, China; duokai305@126.com; 3College of Food and Bioengineering, Henan University of Science and Technology, Luoyang 471023, China; 4College of Food Science, Northeast Agricultural University, Harbin 150030, China; hanjianchun@hotmail.com; 5Heilongjiang Green Food Science Research Institute, Harbin 150028, China

**Keywords:** *Staphylococcus aureus*, *Amaranthus tricolor* crude extract, antimicrobial activity, mechanism of action, cooked pork

## Abstract

*Amaranthus tricolor* has been reported to contain some antimicrobial compounds, such as alkaloids, polyphenols, and terpenoids. However, its effect on *Staphylococcus aureus* has been less well researched. Therefore, this study was designed to evaluate the antimicrobial activity and possible mechanism of action of the *Amaranthus tricolor* crude extract (ATCE) against *S. aureus* and potential application in cooked meat. The antimicrobial activity against *S. aureus* was assessed by disk diffusion, minimum inhibitory concentration (MIC) determinations, and growth curve. The changes of bacterial membrane potential, intracellular pH (pH_in_), content of bacterial protein and DNA, and cell morphology were measured to indicate its antimicrobial mechanism of action. The effects of different concentrations of ATCE on bacterial counts, pH, and color of lean cooked pork during 6 d storage were assessed. The results showed that the diameter of inhibition zone (DIZ) and MIC of ATCE against *S. aureus* were 12.63 ± 0.34 to 12.94 ± 0.43 mm and 80 mg/mL, respectively. The mechanism of action of ATCE against *S. aureus* was associated with cell membrane depolarization, reduction of pH_in_, decrease of bacterial protein content, cleavage of cell DNA, and leakage of cytoplasm. Besides, ATCE resulted in a reduction of 1.02 log CFU/g from 3 log CFU/g in *S. aureus*-inoculated lean cooked pork. The pH values of lean cooked pork treated with ATCE did not show significant changes as the storage time increased, but there was a slight and significant decrease seen with the application of 1 and 2 MIC of ATCE. After treating with ATCE, the color of lean cooked pork showed less lightness (L*), more redness (a∗), similar yellowness (b*), stronger chroma (C*), and weaker hue angle (h*) during 6 days of storage. Therefore, these findings indicate that ATCE has antimicrobial activities against *S. aureus* and possesses latent energy to become a natural preservative to maintain the quality of lean cooked pork.

## 1. Introduction

*Staphylococcus aureus* is a Gram-positive bacterium belonging to the *Firmicutes* family, and is one of the most frequent foodborne pathogenic bacteria [1,2]. It is also one of the most significant zoonotic pathogens and has recently presented a resistance tendency in nosocomial infections [2]. It can survive in a wide range of environments, such as temperatures from 7 to 48.5 °C, pHs between 4.0 and 10.0, and high salt concentration as well as stressful environments, and is easily spread through the air or causes contamination via contact [3,4]. These characteristics contribute to the growth and reproduction of *S. aureus*, and increase the contamination rate [1]. In recent years, outbreaks of foodborne diseases due to the contamination of *S. aureus* were reported often. For example, the European Union reported that 35 outbreaks led to 777 cases, and an estimated 241,000 illnesses strike Americans every year [3,4]. Consumers may be exposed to the risk of *S. aureus* contamination from the processing or manufacturing of cooked meat, dairy, eggs, beans, aquatic products, fresh vegetables, and ready-to-eat foods [2,4,5]. Therefore, some measures (e.g., synthetic preservatives) were taken to control *S. aureus* contamination to ensure food safety, but it is still a problem worthy of attention in the food industry.

A variety of synthetic preservatives (e.g., sodium benzoate and potassium sorbate) are used to inhibit the growth of bacteria in order to ensure food safety. The problem about the safety of synthetic preservatives used in food as well as the effectiveness of natural preservatives is currently in the limelight, and due to their nature and relative safety natural extracts are favored by consumers, which can hopefully be used as alternative food preservatives to inhibit the growth of pathogenic bacteria [6]. Some plant extracts, such as *Thymus vulgaris*, *Origanum vulgare*, *Rosmarinus officinalis,* and Cinnamon extracts, have been demonstrated to exhibit effective antimicrobial activity against *S. aureus* [7,8,9], in which the active ingredients include thymol, carvacrol, camphor, and cinnamaldehyde, respectively. On the other hand, research has shown that phenols and oxygenated terpenoids extracted from plants were the primary effective constituents exerting antimicrobial activity [9]. Trombetta et al. [10] have demonstrated that terpenoids could increase membrane permeability by perturbing the fatty acid composition of the bacterial cell membrane, leading to the leakage of cellular contents. Similarly, the antimicrobial activity of phenols also resulted in structural and functional damages of cell membranes [9].

*Amaranthus* spp. crude extract is a natural substance containing alkaloids (betacyanins and betaxanthin), polyphenols (flavonoids, steroids, catechuic acid, and tannins), terpenoids (cerasinone and norecasantalic acid) and saponins [11]. *Amaranthus viridis*, *Amaranthus hybridus*, *Amaranthus spinosus,* and *Amaranthus caudatus* have been demonstrated to have broad-spectrum anti-bacterial activity [12,13,14]. Amornrit and Santiyanont [14] reported that the leaves extract of *Amaranthus tricolor* showed promising antioxidant properties. In another study, it was found that *Amaranthus tricolor* has a higher phenolic content and antioxidant capacity than *Amaranthus viridis* [15]. Therefore, there is substantial evidence that *Amaranthus tricolor* crude extract (ATCE) has antimicrobial activity, however, few researchers have reported the antimicrobial activity of ATCE against *S. aureus.*

In this study, we aimed to evaluate the antimicrobial activity of ATCE against *S. aureus*, and to elucidate the possible mechanism of action by studying the changes in cell membrane potential, intracellular pH (pH_in_), content of bacterial protein and DNA, and cell morphology after treatment with ATCE. Moreover, the antimicrobial application of ATCE in lean cooked meat against *S. aureus* was also assessed by investigating the growth of *S. aureus*, and the value of pH and color change of lean cooked meat were assessed during storage.

## 2. Materials and Methods

### 2.1. Bacteria and Culture Conditions

*S. aureus* ATCC 13,565 was obtained from American Type Culture Collection (ATCC, Manassas, VA, USA) and the six other *S. aureus* strains (KLDS-sa-1,2; KLDS-sa-3,4; KLDS-sa-5; KLDS-sa-6) were respectively isolated from fresh pork, raw milk, fresh beef, and fresh chicken in Harbin, China. All strains were stored in Luria-Bertani (LB) broth with 20% glycerol (*v*/*v*) at −80 °C. The strains were cultured in LB broth medium at 37 °C for 24 h, and then streaked onto Baird-Parker agar base (BP) plates to be incubated at 37 °C for 24 h. Next, a loopful of each strain was selected and inoculated into LB broth at 37 °C for 20 h to get the pure *S. aureus* cultures for experiment. All isolates were used to assess disk diffusion and minimum inhibitory concentration (MIC), whereas *S. aureus* ATCC 13565 was used to analyze the growth curve, the possible mechanism of action, and application.

### 2.2. Preparation of Amaranthus tricolor Crude Extracts

*Amaranthus tricolor* samples were obtained from a local market of Xiang Fang district, Harbin, Heilongjiang province in China in June 2018. The picked intact leaves of the plants were washed with tap water to remove dirt and aired at 25 °C in the laboratory, then dried at 40 °C and ground into powder using a mechanical grinder (Yongkang boou machinery co., Ltd., Zhejiang, China). Approximately 50 g of the powder was extracted in 1000 mL of 70% (*v*/*v*) ethanol at 40 °C for 2 h. The extracting solution was centrifuged at 2000 × *g* for 10 min (Shanghai Centrifuge Institute Co., Ltd., Shanghai, China) to remove dregs. The supernatant was concentrated in a rotary evaporator at 40 °C until ethanol was removed and then the concentrate was stored at −80 °C to freeze and lyophilize to yield the dried powder. 

### 2.3. Disk Diffusion Assay

The antimicrobial activity of ATCE was investigated using the disk diffusion method [16]. One-hundred microliters of bacterial suspension (approximately 10^6^ CFU/mL) was spread evenly on the solid Luria-Bertani broth containing 1% agar. The filter paper disks with 8 mm diameter were placed on each plate and added to 200 µL of ATCE (320 mg/mL). The plates were incubated at 37 °C for 24 h and the test was performed with three replications. The negative control was same microliters of sterilized water. The diameter (in millimeters) of the inhibition zone (DIZ) in which the colonies did not grow was measured to visually assess the antimicrobial effect of ATCE.

### 2.4. Determination of MIC 

The MIC of ATCE against *S. aureus* strains was determined using the agar dilution method recommended by a previous report [17]. The different concentrations of ATCE (10, 20, 40, 80, 160, 320, and 640 mg/mL) were thoroughly mixed with TSA (approximately 50 °C) in sterile 24-well plates, and 0.1 mg/mL ampicillin was used as the positive control. After hardening, 2 µL of tested *S. aureus* strain (about 10^6^ CFU/mL) was transferred into the culture medium and allowed to dry. Then the 24-well plates were incubated at 37 °C for 24 h. The MIC of ATCE against *S. aureus* was considered as the lowest concentration at which the visible growth of *S. aureus* was inhibited completely. 

### 2.5. Growth Curve of S. aureus ATCC 13565

The growth curves of *S. aureus* ATCC 13565 were tested according to the method described by Shi et al. [18]. *S. aureus* ATCC 13565 was grown in LB broth until an OD_600nm_ value of 0.2 was detected for the bacterial suspension. Then, 125 µL of the bacterial suspension was added to each well in 96-well microtiter plates. The ATCE was added to the cultures and the final concentrations were adjusted to 0, 0.5, 1, 1.5, and 2 MIC, respectively. After being further cultured at 37 °C, the cell growth was monitored using a multimode plate reader at 600 nm (Tecan, Infinite^TM^ M200 PRO, Männedorf, Switzerland) at 2 h intervals.

### 2.6. Measurement of Membrane Potential

The membrane potential of *S. aureus* ATCC 13565 with different ATCE treatments was measured according to the previous reports by Fei et al. [19]. A membrane-potential-sensitive fluorescent probe, bis-(1,3-dibutylbarbituric acid) trimethine oxonol (DiBAC4(3); Beijing Solarbio Science and Technology Co. Ltd., Beijing, China) and 125 μL of bacteria solution (approximately 10^7^ CFU/mL) were added in a black and opaque 96-well plate (Corning Institute, New York City, New York State, USA) and incubated at 30 °C for 30 min. Then, *S. aureus* ATCC 13565 were treated with 0, 1, and 2 MIC of ATCE that were added in the 96-well plate. The fluorescence intensity of each well was measured using a multifunctional microplate reader at 492 nm excitation wavelengths and 515 nm emission wavelengths. The result of membrane potential was displayed by the value of relative fluorescence units (RFUs) as an indicator of signal intensity.

### 2.7. Determination of Intracellular pH (pH_in_)

The changes in *S. aureus* ATCC 13565 cells’ intracellular pH (pH_in_) were assayed according to previous study with slight modifications [18]. Approximately 10^8^ CFU/mL of *S. aureus* ATCC 13565 bacterial suspension were incubated at 37 °C for 30 min in the presence of 1.0 µM carboxyfluorescein diacetate succinimidyl ester (CFDA SE) dye that was used as the pH_in_ fluorescent probe. The cells were centrifuged at 11,200 × *g* for 5 min and resuspended in sterile normal saline (NS). The bacterial suspension was added in 10 mM glucose solution and incubated for 30 min at 37 °C, followed by centrifuging at 11,200 × g for 5 min, washed twice and resuspended in NS. Equal volumes (125 μL) of bacteria solution and 0, 1, and 2 MIC of ATCE were added in a black and opaque 96-well plate. Fluorescence intensities were detected by a microplate reader after treatment for 20 min (Tecan, Infinite^TM^ M200 PRO, Männedorf, Switzerland). The measurement system was set to excitation wavelengths of 440 nm and emission wavelengths of 490 nm, and maintained at 25 °C. The pH_in_ was determined as the ratio of the fluorescence signal at the pH-sensitive wavelength (490 nm) and that at the pH-insensitive wavelength (440 nm). The ultimate fluorescences of samples were calculated by deducting the fluorescence of cell-free controls.

### 2.8. Sodium Dodecyl Sulfate-Polyacrylamide Gel Electrophoresis (SDS-PAGE)

The effect of ATCE on the bacterial protein of *S. aureus* ATCC 13565 was analyzed using SDS-PAGE according to Chen et al. [20]. Approximately 10^7^ CFU/mL of *S. aureus* ATCC 13,65 bacterial suspension was treated with 1 and 2 MIC of ATCE for 3, 6, 9, and 12 h at 37 °C followed by withdrawal of bacterial suspensions and centrifugation (8000 × *g*) at 4 °C for 10 min to prepare the supernatants. Untreated *S. aureus* ATCC 13565 cells were analyzed with the same processing as a negative control. The mixture of supernatants (10 μL) and SDS-PAGE loading buffer (5 μL) was then heated at 95 °C for 10 min and the bacterial proteins was separated by SDS-PAGE using a 5% stacking gel and a 15% separating gel. Finally, the protein bands were stained with Coomassie brilliant blue R-250 (Beijing Solarbio Sciences and Technology Co. Ltd., Beijing, China). The images were taken with an HP scanner (HP 1000, Silicon Valley, California, USA).

### 2.9. Agarose Gel Electrophoresis for DNA Fragmentation

The DNA fragmentation of *S. aureus* ATCC 13565 was determined using agarose gel electrophoresis as mentioned in previous research [21]. Approximately 10^8^ CFU/mL of *S. aureus* ATCC 13565 cells was treated with 0, 1, and 2 MIC of ATCE at 37 °C for 2, 4, and 10 h. The genomic DNA was extracted according to the instructions of a bacterial genomic DNA extraction kit (Tiangen Biotech Co., Ltd., Beijing, China). The extracted DNA samples were electrophoresed using agarose gel (1.5%) at 100 V for 30 min and the gels were stained for 15 min using 10 mg/mL of ethidium bromide. Finally, the gels were visualized by a gel imaging system (Bio-Rad, Hercules, California, USA).

### 2.10. Transmission Electron Microscopy

The cellular morphology of *S. aureus* ATCC 13565 cells with different ATCE treatments was observed using a transmission electron microscope (Hitachi, Tokyo, Japan) according to previous reports [17]. After treatments with 0, 1, and 2 MIC of ATCE for 4 h, the working *S. aureus* ATCC 13565 cells were collected by centrifugation at 7000 × *g* for 10 min and fixed in 0.1 M sodium phosphate buffer containing 2.5% glutaraldehyde at 4 °C for 2 h and then washed three times with NS. After dehydrating with 50%, 70%, 90%, and 100% ethanol for 10 min, the cells were embedded in Epon Lx-112 (Ladd Research, Williston, North Carolina, USA). Then, the samples were cut into sections of 50 to 60 nm and stained using double staining with uranyl acetate and lead citrate. Finally, the cell morphology of samples was observed under the TEM.

### 2.11. Application of ATCE in Cooked Meat and Microbiological Analysis

Fresh lean pork was purchased from the local market and cut into cube blocks (approximately 2 × 2 × 2 cm^3^). Then the cube blocks were sterilization for 20 min at 121 °C to obtain the sterile lean cooked pork samples. The sterile samples were treated with 0, 1, and 2 MIC of ATCE for 30 min, respectively. Approximately 10^3^ CFU/mL of *S. aureus* ATCC 13565 cells was inoculated into the samples treated with ATCE. The bacterial suspension was mixed with each sample for 15 s using a sterile applicator stick. Then, the inoculated samples were sealed and stored using aseptic bags at 4 °C for 6 days. On days 0, 3, and 6, the samples (10 g) were taken out in turn and mashed (Stomacher 400 Laboratory Blender, Seward, Worthington, UK) on medium power mode for 30 s in 0.1% sterile NS (90 mL). The suspension liquid samples were appropriately diluted by NS. Finally, the number of *S. aureus* ATCC 13565 on lean cooked pork was enumerated on Baird-Parker (BP, Difco Labs, Detroit, Michigan, USA) plates by spreading 100 μL of the sample dilution. 

### 2.12. The pH Value Measurement

The cooked pork samples were treated with 0, 1, and 2 MIC of ATCE for 30 min and then sealed and stored using aseptic bags at 4 °C for 6 days. On days 0, 3, and 6, the treated samples (10 g) were homogenized in 90 mL distilled water using an Ultra Turrax T25 homogenizer (Janke and Kunkel, IKA-Labortechnik, GmbH and Co., Staufen, Germany). The pH of the samples was measured using a portable pH meter (Radiometer, Copenhagen, Denmark).

### 2.13. Color Determination

The pretreatment process for cooked pork samples was the same as for pH measurement in Section 2.12. The surface color of cooked pork samples was measured using a CR-300 Chroma Meter (Minolta Co., Osaka, Japan) with calibration using a white tile according to the CIE color space system [22]. The L*, a*, and b* values represent lightness, redness, and yellowness, respectively. In addition, the chroma value (C*) and hue angle value (h*) were generate according to the following formulas from previous study [23].

(1)C*=[(a*2+b*2)12]

(2)h*=[arctangent (b*a*) ]

### 2.14. Statistical Analysis

The data obtained in this study were described by means and standard deviations. The date of the disk diffusion assay, membrane potentials, pH_in_ values, the bacterial survival counts, and the values of pH and color determination were analyzed using both analysis of variance (ANOVA) and Duncan’s test. The two methods were used to determine the significance level and get a more rigorous data analysis. In addition, SPSS 22.0 software (SPSS Inc., Chicago, IL, USA) was used to conduct the ANOVA. Significant differences among the means (*p* < 0.05) was confirmed by Duncan’s multiple range test. All experiments were carried out in triplicate.

## 3. Results

### 3.1. Antimicrobial Effects of ATCE on S. aureus

The antimicrobial effect of ATCE against *S. aureus* was shown in Table 1. The DIZs of seven *S. aureus* strains ranged from 12.63 ± 0.34 to 12.94 ± 0.43 mm in the presence of 320 mg/mL of ATCE and the DIZ values of ATCE against *S. aureus* strains showed no significant difference (*p* > 0.05). Bacterial colonies of seven *S. aureus* did not grow when treated with 80 mg/mL of ATCE. Therefore, the MIC of ATCE against seven *S. aureus* strains was determined to be 80 mg/mL.

### 3.2. Growth Curve of S. aureus ATCC 13565

The antimicrobial activity of ATCE against *S. aureus* ATCC 13565 was evaluated by growth curve analysis, at different concentrations ranging from 0.5 MIC to 2 MIC. The results indicated that effective inhibition of growth was achieved after 10 h by ATCE at 1, 1.5, and 2 MIC. In addition, ATCE at 0.5 MIC slightly inhibited *S. aureus* ATCC 13565 in comparison with the control (Figure 1).

### 3.3. Changes in Membrane Potential

The changes of membrane potential can be reflected by observing the fluorescence intensity in cells, and the increase (decrease) of fluorescence intensity indicated cell membrane depolarization (hyperpolarization). As shown in Figure 2, compared with the untreated group, the fluorescence intensities of *S. aureus* ATCC 13565 cells treated with ATCE were significantly increased (*p* < 0.05). This indicates that ATCE can arouse depolarization in *S. aureus* ATCC 13565 cells. Moreover, the cells treated with 1 and 2 MIC of ATCE showed no difference in fluorescence intensity (*p* > 0.05).

### 3.4. Changes in pH_in_

A clear change in pH_in_ appeared after adding ATCE (Figure 3). After treatment with 1 MIC of ATCE, the values of *S. aureus* ATCC 13565 cell pH_in_ decreased from 8.02 ± 0.77 to 5.44 ± 1.07. In addition, the values of *S. aureus* ATCC 13565 cell pH_in_ decreased significantly (*p* < 0.05) from 8.02 ± 0.77 to 4.78 ± 0.72 at 2 MIC of ATCE, where the reduction was greater than that for 1 MIC of ATCE, but not significantly different (*p* > 0.05). This also suggests that ATCE is bactericidal to *S. aureus* ATCC 13565 when the concentration is greater than or equal to 2 MIC.

### 3.5. SDS-PAGE Analysis

The SDS-PAGE images indicate that the gel bands of *S. aureus* ATCC 13565 treated with 1 and 2 MIC of ATCE gradually faded as time went on (Figure 4). The protein bands became much weaker and disappeared after treatment with 2 MIC of ATCE for 12 h, suggesting ATCE has a bactericidal effect on *S. aureus* ATCC 13565. As ATCE treatment concentration increased, the bands became weaker and the protein contents reduced.

### 3.6. DNA Cleavage Analysis

The results of teh genomic DNA electrophoretogram of bacterial cells treated with ATCE are shown in Figure 5. The DNA bands of *S. aureus* ATCC 13565 after exposure to 1 and 2 MIC of ATCE became faint or even disappeared as the concentration of ATCE and time increased, compared to the untreated samples. The DNA bands disappeared when *S. aureus* ATCC 13565 was treated with 2 MIC of ATCE for 10 h. This suggests that ATCE has bactericidal function on *S. aureus* ATCC 13565 when the concentration is greater than or equal to 2 MIC.

### 3.7. Transmission Electron Microscope Observation

TEM was used to observe the cell morphology of *S. aureus* ATCC 13565 (Figure 6). Untreated cells of *S. aureus* ATCC 13565 maintained normal morphology and intact cell structure (Figure 6A). The cytoplasmic membrane was gradually separated from the cell wall and the intracellular components leaked after treatment with 1 MIC of ATCE (Figure 6B). Meanwhile, the degree of cell deformation and cytoplasmic leakage obviously increased when the *S. aureus* ATCC 13565 cells were treated with 2 MIC of ATCE (Figure 6C).

### 3.8. Antimicrobial Effects of ATCE

*Amaranthus tricolor* crude extract (ATCE) at 1 and 2 MIC slightly reduced the numbers of *S. aureus* ATCC 13565 in inoculated lean cooked pork in the first 3 days of storage compared to the initial populations (Figure 7), and there was no significant difference between the results of these two concentrations (*p* > 0.05). The numbers of *S. aureus* ATCC 13565 treated with 2 MIC was significantly lower than in the sample treated with 1 MIC after 6 days (*p* < 0.05).

### 3.9. Effects of ATCE on pH Change

The pH values of lean cooked pork over the 6 days of storage did not show significant changes at 1 and 2 MIC of ATCE (*p* > 0.05). However, after being treated with different concentrations of ATCE, the pH values of lean cooked pork treated with 1 and 2 MIC of ATCE were slightly decreased (*p* < 0.05) compared to the control during the same storage time (Table 2).

### 3.10. Effects of ATCE on Color Change

The effects of ATCE treatment on the color changes of lean cooked pork during storage are shown in Table 3. The L* value of lean cooked pork was decreased by treatment with ATCE. The addition of ATCE increased the a* value of the cooked pork as the extract concentration of ATCE increased (*p* < 0.05). The b* values of samples treated or not treated with ATCE displayed no significant differences during storage (*p* > 0.05). The C* values increased slightly as the storage time increased and significantly increased (*p* < 0.05) when the ATCE treatment concentration increased at same storage time. The h* value of cooked pork that was treated with the same ATCE concentration did not significantly change during storage. However, the h* value declined as the concentration of ACTE increased and the number of storage days was the same.

## 4. Discussion

Phytochemicals like alkaloids, polyphenols, terpenoids, and saponins have been established as the active constituents present in *Amaranthus* spp., and the plant extracts have been shown as effective against foodborne pathogens [11]. Some plants of *Amaranthus* spp. have also demonstrated general action against *S. aureus* strains. Bulbul et al. [12] and Tahir and Khan [13] found that the DIZ was 14.5 and 13.80 mm, respectively, when strains of *S. aureus* were treated with *n*-hexane extracts of *Amaranthus spinosus* and *Amaranthus viridis*. Maiyo et al. [24] reported antimicrobial effects of *Amaranthus caudatus* against *S. aureus* strains, with an MIC of 155.6 mg/mL. In addition, some polyphenol extracts such as oregano extracts (including thymol, carvacrol, and eugenol) and olive oil polyphenol extract have been determined to possess antimicrobial activity against *S. aureus* with MICs of 10 mg/mL and 1.25 mg/mL, respectively [7,8]. In this study, the DIZ and MIC of ATCE against *S. aureus* ATCC 13565 strains were 12.82 ± 0.50 mm and 80 mg/mL, which indicated that antimicrobial activity of ATCE was effective. The growth curve showed that ATCE inhibited the growth of *S. aureus* ATCC 13565, and the lag phase and specific growth rate in the exponential phase became longer and lower with the increase of ATCE concentrations, which was similar with the growth of *Cronobacter sakazakii* treated with citral [18]. This result indicated that the antimicrobial activity of ATCE was worth confirming.

It is essential for life of the all bacteria to maintain a membrane potential, which is used for producing energy, driving active transport of molecules across the membrane, empowering motility (flagella), and preventing the cytoplasm from equilibrating with the environment [25]. Membrane changes (depolarization and hyperpolarization) have been suggested to be one of the primary indicators of injured bacteria [17]. In this study, the results showed that *S. aureus* ATCC 13565 displayed cell membrane depolarization after treatment with ATCE, which occurs primarily due to the release of K^+^ or K^+^ with several other ions [18,25]. Similar results were also reported in the cells of *Listeria monocytogenes* exposed to olive oil polyphenol extract and *Cronobacter sakazakii* treated with citral [18,21]. In addition, the occurence of cell membrane hyperpolarization was associated with change in pH and leakage of K^+^ [26].

Maintenance of pH_in_ homeostasis is considered to be critical to keep the correct DNA transcription and protein synthesis of the bacteria cell [27]. Cells with an intact membrane can maintain their internal pH through ion channels and pumps when the outside pH undergoes gentle change [18]. In this study, ATCE induced a decrease in the pH_in_ of *S. aureus* ATCC 13565 strains, indicating that membrane damage occurred. Similarly, Lambert et al. [28] demonstrated that thymol and carvacrol reduced the pH_in_ of *S. aureus*. Shi et al. [29] found that reductions of the internal pH of *C. sakazakii* strains were detected after treatments with lipoic acid. Therefore, the damage of cell membrane integrity led to the reduction of pH_in_, which is considered as one of the important antimicrobial pathways of natural substances.

As the basic substance of life, protein plays an important part in the activity and physiological function of bacterial cells [19]. The contents of protein in *S. aureus* ATCC 13565 cells was decreased with treatment of antimicrobial agents, such as sugarcane bagasse extract and sugar beet molasses polyphenols [20,28]. In this study, the SDS-PAGE results also confirmed this. These results proved that natural extracts maybe have interacted with protein, increased the membrane permeability, and disturbed the synthesis of bacterial proteins. The membrane structure of the cell was destroyed by ATCE, resulting in the leakage of intracellular soluble contents. Therefore, the contents of bacterial protein reduced and the protein bands became weaker or not apparent as the treatment concentration of ATCE increased.

DNA carrying genetic information is in charge of bacterial growth, development, and inheritance [30]. After a treatment with ATCE, the bacterial DNA bands became fainter or even disappeared, resulting in DNA fragmentation. Moreover, this destructive effect was related to the concentration of ATCE, which was in step with the results in previous studies that cajanol inhibits the growth of *S. aureus* by DNA cleavage [31]. It can be supposed that the possible mechanism of action is that ATCE destroys the bacterial cell membrane and increases its permeability, resulting in the leakage of DNA or promoting the entrance of ATCE in bacterial cells and interaction with DNA, disturbing gene expression and thereby causing bacterial cell death.

TEM observation is powerful tool to better observe the cell morphology of natural products’ action on bacterial cells [17]. After treatment with ATCE, the cell morphology of *S. aureus* ATCC 13565 was obviously seriously damaged and there was cytoplasmic leakage as the cell membrane permeability increased, breaking the cell homeostasis. Similar results have been found that sugar beet molasses polyphenols and sugarcane bagasse extract caused *S. aureus* cell deformation and cytoplasmic leakage [20,28]. Therefore, the visualization of membrane damage under TEM confirmed the increase of cell membrane permeability and the destruction of membrane structures by natural extracts, leading to cell membrane depolarization, decreases in pH_in_, leakage of cytoplasm, bacterial protein reduction and DNA cleavage. The results demonstrated a valid antimicrobial effect against *S. aureus* strains. 

Cooked meat products are generally regarded as safe, but can be contaminated by foodborne pathogens such as *S. aureus* during production, manufacturing, distribution, and preparation [31,32]. More importantly, cooked meat if contaminated by foodborne pathogens not only has spoiled flavor, aroma, taste, nutritional value, and overall quality, but also threatens human health and leads to illness and death [8]. In this study, ATCE could inhibit the growth of *S. aureus* and was found to injure cell membranes and depolarize bacteria cells, reduce pH_in_, and change cell morphology, as well as reduce the bacterial protein content and cleave DNA to prevent the growth of *S. aureus*. Thus, ATCE possesses the potential to become a food additive due to its antimicrobial activities and can be used to improve the shelf-life and the safety of cooked meat. In addition, the ATCE action was proved by decreasing the numbers of *S. aureus* ATCC 13565 in inoculated lean cooked pork in this study. A similar study indicated that the grape seed extract inhibited the growth of *E. coli* O157:H7 and *Salmonella typhimurium* in cooked beef [22], showing that the natural extracts effectively inhibited microbial growth.

Natural antimicrobial systems could improve the stability and safety of cooked meats, and could also be influenced by many factors, such as the food components, pH, as well as storage temperature [33]. Some studies have reported that the pH of cooked pork treated by plant extracts was within the range of 5.92–6.07 and concluded that the pH of the pork was unaffected by the addition of the extracts [5,33]. This phenomenon was consistent with the results of this study, where the pH values of lean cooked pork treated with different ATCE concentrations was decreased during the same storage time, but there were no significant changes with increased storage time. In brief, the ATCE had little influence on the pH value of lean cooked pork.

The color and color uniformity of cooked meats are the crucial conditions for consumer acceptance. The meat color can be simply reflected by lightness (L*) and redness (a*), but the yellowness (b*) is not typically or intuitively related to meat [34]. The C* value indicates the hue intensity of the product, and larger values represent a greater saturation of the principle hue [35]. The h* value indicates color changes during shelf life and larger values indicate less redness and a more well-done cooked color [35]. ATCE increased the L* and a* values of the lean cooked pork as the concentration increased. The slight increase of L* values and the pronounced increase of a* values of lean cooked pork was possibly due to the red-brown colour of the ATCE extract. Ifesan et al. [5] found that the addition of the *Eleutherine americana* crude extract led to an increase in the redness values of the pork but did not change the L* values. Moreover, the color (L* and a*) change of lean cooked pork treated with ATCE may result from the antioxidative effects of natural substances (e.g., alkaloids, polyphenols, and terpenoids). The increased C* value of cooked pork treated with ATCE showed the saturation enhancement of the principle hue. The decreased h* value indicated a greater redness of the cooked meat. The results all demonstrate that ATCE was beneficial to the color of lean cooked pork. Color is an important visual cue, and an attractive food color can improve consumer acceptance of food and increase the desire to consume [36].

## 5. Conclusions

This study indicates that ATCE possesses antimicrobial activity against *S. aureus* and suggests that ATCE has a bactericidal function against *S. aureus* ATCC 13565 when the concentration is greater than or equal to 2 MIC. The antimicrobial action was associated with cell membrane depolarization, decrease of pH_in_, bacterial protein reduction, DNA cleavage, and changed cell morphology. Moreover, its application to cooked pork preservation manifested as decreased numbers of *S. aureus* ATCC 13565 in inoculated lean cooked pork. The pH values of lean cooked pork treated with ATCE did not show significant changes as the storage time increased and saw a slight decrease during the same time period as the concentration was increased. The color of lean cooked pork treated with ATCE showed decreased L* values, increased a* values, no significant changes to b* values, increased C* values, and decreased h* values during storage. These findings indicate that ATCE has the potential ability to control the growth of *S. aureus* in the food industry. However, ATCE dosage optimization, possible interactions between food components and natural extracts, and security problems need to be researched in the future.

## Figures and Tables

**Figure 1 foods-09-00359-f001:**
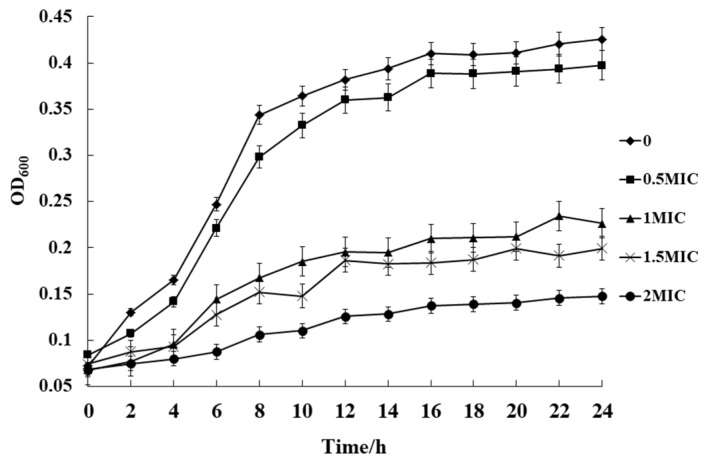
Growth curves of *S. aureus* ATCC 13565 cultured in Luria-Bertani (LB containing with 0, 0.5, 1, 1.5, and 2 MIC of ATCE. Error bars denote standard deviation (*n* = 3).

**Figure 2 foods-09-00359-f002:**
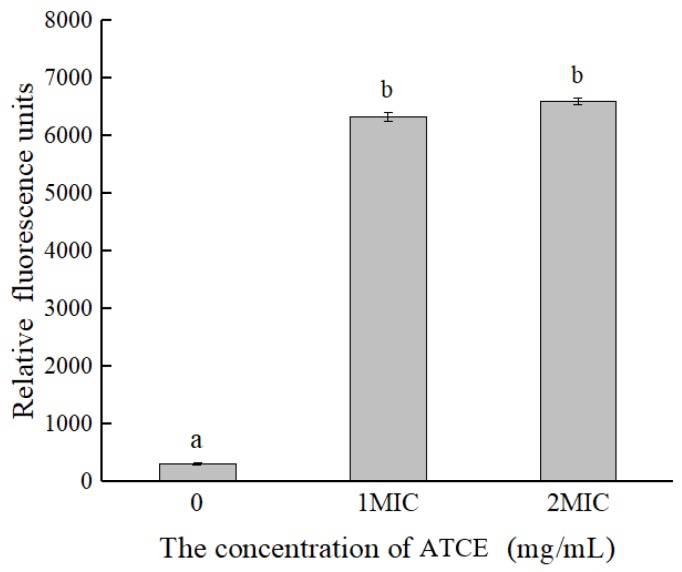
Differences in membrane potentials of *S. aureus* ATCC 13565 after treatment with ATCE at 0, 1, and 2 MIC. Values denote the means of triplicate measurements. Error bars represent standard deviation (*n* = 3). Different lowercase letters (a,b) represent significant differences (*p* < 0.05).

**Figure 3 foods-09-00359-f003:**
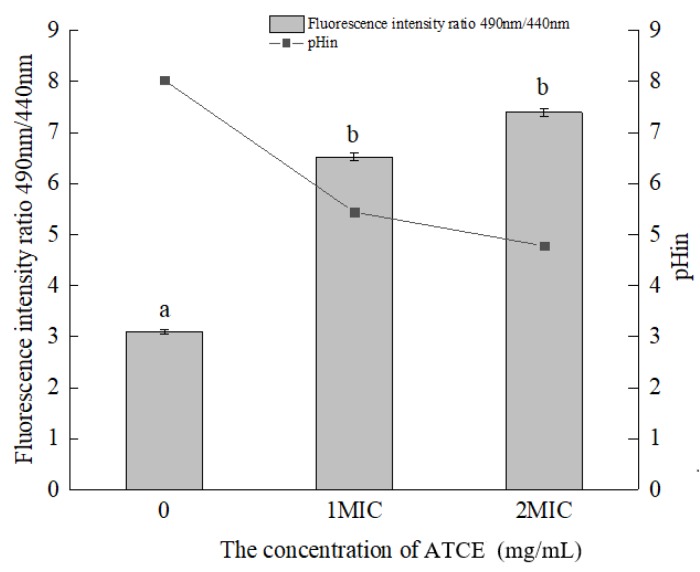
Differences in the intracellular pH (pHin) of *S. aureus* ATCC 13565 after treatment with ATCE at 0, 1, and 2 MIC. Values denote the means of triplicate measurements. Error bars represent standard deviation (*n* = 3). Different lowercase letters (a, b) represent significant differences (*p* < 0.05).

**Figure 4 foods-09-00359-f004:**
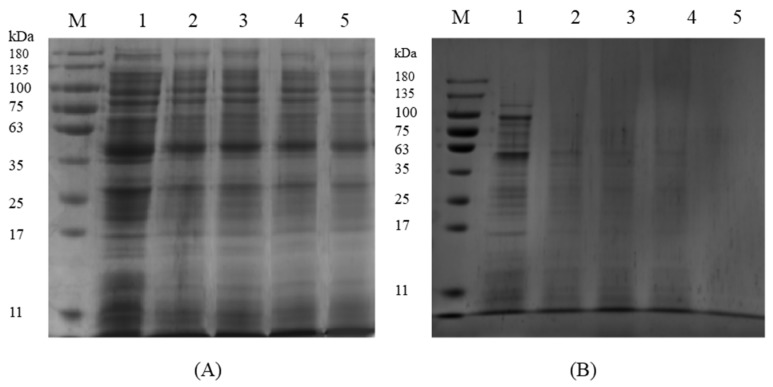
SDS-PAGE analysis of *S. aureus* ATCC 13565 proteins after treatment with ATCE at (**A**) 1 MIC and (**B**) 2 MIC. Lane M: marker. Lanes 1: control; Lanes 2, 3, 4, and 5: samples treated for 3, 6, 9, and 12 h, respectively.

**Figure 5 foods-09-00359-f005:**
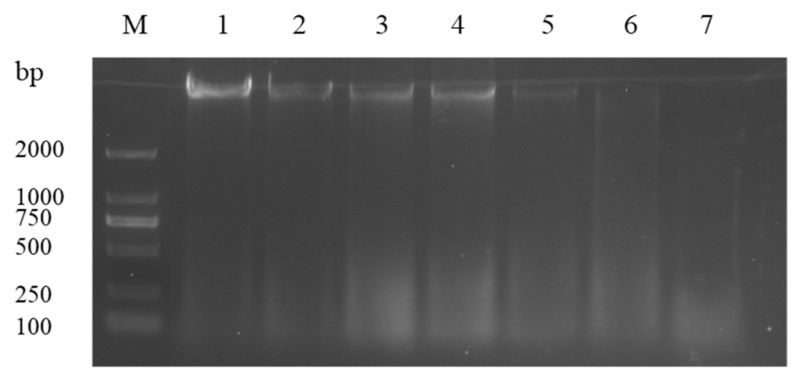
DNA cleavage activity of *S. aureus* ATCC 13565 strains after treating with ATCE at 1 MIC and 2 MIC. Lane M: marker. Lane 1: control group. Lanes 2, 3, and 4: treated with 1 MIC of ATCE for 2, 4, and 10 h, respectively. Lanes 5, 6, and 7: treated with 2 MIC of ATCE for 2, 4, and 10 h, respectively.

**Figure 6 foods-09-00359-f006:**
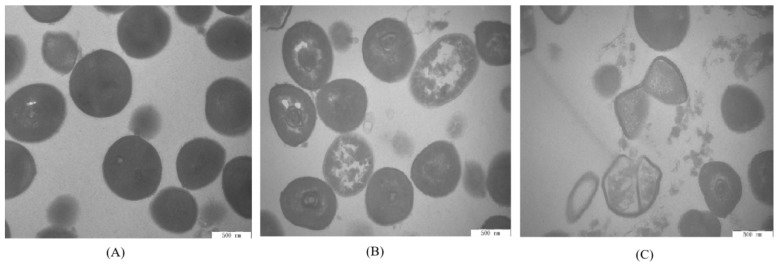
TEM images of *S. aureus* ATCC 13565 strains (40,000 ×) (**A**) untreated for 4 h, (**B**) treated with 1 MIC of ATCE for 4 h, and (**C**) treated with 2 MIC of ATCE for 4 h.

**Figure 7 foods-09-00359-f007:**
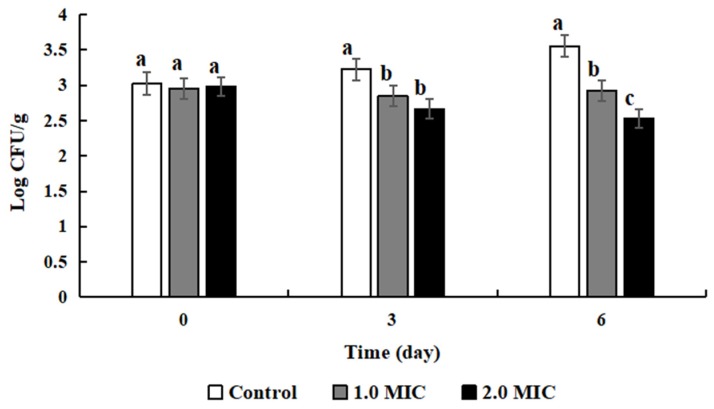
Antibacterial effect of ATCE on *S. aureus* ATCC 13565 inoculated into sterile lean cooked pork during storage. Values denote the means of triplicate measurements. Error bars represent standard deviation (*n* = 3). Different letters represent significant differences (*p* < 0.05).

**Table 1 foods-09-00359-t001:** Antimicrobial activity of *Amaranthus tricolor* crude extract (ATCE) against all six *Staphylococcus aureus* strains.

Strain	DIZ (mm)	MIC (mg/mL)
ATCC 13565	12.82 ± 0.50 ^a^	80
KLDS-sa-1	12.94 ± 0.43 ^a^	80
KLDS-sa-2	12.71 ± 0.85 ^a^	80
KLDS-sa-3	12.78 ± 0.51 ^a^	80
KLDS-sa-4	12.86 ± 0.72 ^a^	80
KLDS-sa-5	12.89 ± 0.57 ^a^	80
KLDS-sa-6	12.63 ± 0.34 ^a^	80

^a^ Column values with different lowercase letters are significantly different (*p* < 0.05). DIZ: diameter of the inhibition zone; MIC: minimum inhibitory concentration.

**Table 2 foods-09-00359-t002:** The pH values of cooked pork with ATCE during storage for 6 days.

Storage Time (Days)	Treatments (Cooked Meat + Extract)		
	Control	1 MIC	2 MIC
0	6.04 ± 0.02 ^aA^	5.99 ± 0.02 ^bA^	5.89 ± 0.01 ^cA^
3	6.05 ± 0.02 ^aA^	5.98 ± 0.02 ^bA^	5.90 ± 0.01 ^cA^
6	6.05 ± 0.02 ^aA^	5.99 ± 0.02 ^bA^	5.88 ± 0.02 ^cA^

All the values are the means of three replicated measurements ± standard deviation (SD). Values with different superscript letters (a–c) within a row in a row are significantly different (*p* < 0.05). Values with the same superscript letters (A) within a row in a row are not significantly different (*p* > 0.05).

**Table 3 foods-09-00359-t003:** Color changes of the cooked pork treated with ATCE during storage.

Storage Time (Days)	Treatments (Cooked Meat + Extract)		
	Control	1 MIC	2 MIC
L* value	-	-	-
0	63.70 ± 0.16 ^A^^a^	48.66 ± 0.11 ^B^^b^	42.11 ± 0.34 ^A^^c^
3	65.44 ± 0.04 ^C^^a^	48.50 ± 0.18 ^B^^b^	43.28 ± 0.26 ^B^^c^
6	63.99 ± 0.06 ^B^^a^	47.71 ± 0.20 ^A^^b^	42.61 ± 0.27 ^A^^c^
a* value	-	-	-
0	3.83 ± 0.08 ^Aa^	11.48 ± 0.28 ^A^^b^	14.37 ± 0.25 ^A^^c^
3	3.54 ± 0.09 ^B^^a^	11.89 ± 0.23 ^A^^b^	13.97 ± 0.07 ^A^^c^
6	3.14 ± 0.06 ^Ca^	11.76 ± 0.37 ^A^^b^	13.91 ± 0.34 ^A^^c^
b* value	-	-	-
0	16.71 ± 0.06 ^Aa^	16.15 ± 0.23 ^A^^b^	16.28 ± 0.08 ^Ab^
3	16.67 ± 0.22 ^Ba^	16.16 ± 0.05 ^A^^b^	16.07 ± 0.26 ^Ab^
6	17.37 ± 0.20 ^Aa^	16.52 ± 0.12 ^B^^b^	16.01 ± 0.10 ^Ac^
C* value	-	-	-
0	17.14 ± 0.07 ^Aa^	20.11 ± 0.15 ^Ab^	21.71 ± 0.14 ^Ac^
3	17.03 ± 0.29 ^Aa^	20.20 ± 0.21^Bb^	21.28 ± 0.24 ^Ac^
6	17.65 ± 0.25 ^Ba^	20.27 ± 0.24 ^Bb^	21.21 ± 0.17 ^Bc^
h* value	-	-	-
0	1.34 ± 0.01 ^Aa^	0.95 ± 0.02 ^Ab^	0.83 ± 0.01 ^Ac^
3	1.36 ± 0.00 ^Ba^	0.93 ± 0.01 ^Ab^	0.85 ± 0.01 ^Ac^
6	1.39 ± 0.00 ^Ca^	0.95 ± 0.02 ^Ab^	0.85 ± 0.02 ^Ac^

All the values are the means of three replicated measurements ± standard deviation (SD). Values with different superscript letters (a-c) within a row in a row are significantly different (*p* < 0.05). Values with different superscript letters (A-C) within a row in a row are significantly different (*p* < 0.05).

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
