# Peer review of "Antimicrobial Activity and Mechanism of Action of the Amaranthus tricolor Crude Extract against Staphylococcus aureus and Potential Application in Cooked Meat"

_foods, 2020, doi:10.3390/foods9030359_

Round 1

Reviewer 1 Report

  • Authors need to address most of these comments.
  • Introduction need major rewrite as the current form is just series of sentences with no flow or coherence message being passed.
  • Materials and Methods sections need to be rewritten especially sections 2.1-2.5 and 2.11
  • There is a need of consistency and flow of discussion in the Results and Discussions sections

Author Response

Response to Reviewer 1 Comments

Comments and Suggestions for Authors:

Authors need to address most of these comments.

Point 1: Introduction need major rewrite as the current form is just series of sentences with no flow or coherence message being passed.

Response 1: We thank you very much for your positive comments on our manuscript, and agree with you on the revision of the manuscript. Based on your precious comments, we have carefully revised the introduction section.

Point 2: Materials and Methods sections need to be rewritten especially sections 2.1-2.5 and 2.11.

Response 2: We appreciate your valuable comments and rewrited the sections 2.1-2.5 , 2.11 and others in the maked up manuscript.

Point 3: There is a need of consistency and flow of discussion in the Results and Discussions sections.

Response 3: We thanks very much for your contribution to the improvements to the manuscript. We have revised our manuscript according to your comments carefully.

The other comments in “foods-733227-review”

Point 4: Is there no better wording than this (line 2: “Action Approach”)

Response 4: We agree with and thanks for your doubt. We also considered the similar question when we determined our title and hesitated to use “action approach” or “mechanism of action”. Now, “action approach” has been replaced by “mechanism of action” in revised manuscript.

Point 5: Wording is not correct (line 22: “reveal”)

Response 5: Thanks for your carefulness.“reveal” has been replaced by “evaluate”. P1 line 22 in revised manuscript.

Point 6: Reduced by 1.02 log CFU/g from what amount (line 33: “ATCE resulted in reduction of 1.02 log CFU/g, in number of S. aureus inoculated lean cooked pork.”)

Response 6: Thank you very much. This sentence has been rewritten as “ATCE resulted in reducing by 1.02 log CFU/g from 3 log CFU/g S. aureus inoculated lean cooked pork.”  P1 line 33 in revised manuscript.

Point 7: Revise your wording (line 37: “proved”)

Response 7: Thanks a lot. “proved” has been replaced by “indicated”. P1 line 38 in revised manuscript.

Point 8: What does this mean (line 46: “sodium chloride”)

Response 8: It has been corrected. P2 line 49 in revised manuscript. Many thanks.

Point 9: Check your wording (line 51: “found”)

Response 9: Thank you very much. It has been corrected. P2 line 54 in revised manuscript.

Point 10: This is not clear (line 52: detected due to the deficient hygiene”)

Response 10: Thanks for your reminder. It has been rewritten. P2 lines 54 and 55 in revised manuscript.

Point 11: What technologies and safety systems (line 55: “industry using technologies and safety systems.”)

Response 11: We are really sorry that our unclear expression. The sentence has been rewritten as “Therefore, some measures, such as synthetic preservatives was taken to control S. aureus contamination to ensure food safety, but it is still a problem worthy of attention in food industry.” P2 lines 56-58 in revised manuscript.

Point 12: Such as... (line 56: “A variety of synthetic preservatives”)

Response 12: Many thanks. The example “sodium benzoate and potassium sorbate” has been added. P2 lines 59 in revised manuscript.

Point 13: grammatically not correct (line 61: “has”)

Response 13: We are very sorry for our negligence. “has” has been replaced by “had”. P2 line 65 in revised manuscript.

Point 14: Too general (line 63 and 64: “exhibited antimicrobial activity are phenols, followed by oxygenated terpenoids [9]. Trombetta et al. [10] have”)

Response 14: Thank you very much for your comment. It has been corrected. P2 lines 66-68 in revised manuscript.

Point 15: Would be better to at least mention one of two constituents of those groups ( line 68 and 69: “Amaranthus spp. crude extract is a natural substance contains alkaloids (betaine), polyphenols (flavonoids, steroids,catechuic and tannins), terpenoids and saponins [11].”)

Response 15: Many thanks. It has been mentioned according to your comment. P2 lines 73-75 in revised manuscript.

Point 16: Check the language (line 74: “should have”)

Response 16: Thank you for your carefulness.“should have” bas been replaced by“had”. P2 lines 80 in revised manuscript.

Point 17: ...the value of pH and colour change have nothing to do with AM application ATCE. Need to rewrite it (line 79-81: “Moreover, the antimicrobial application of ATCE in lean cooked meat against S. aureus was also assessed by investigating the the growth of S. aureus, the value of pH and color changes of lean cooked meat.”)

Response 17: Thanks very much for your reminder. The sentence has been rewritten as “Moreover, the antimicrobial application of ATCE in lean cooked meat against S. aureus was also assessed by investigating the the growth of S. aureus, and the value of pH and color change of lean cooked meat were assessed during storage.” P2 lines 86 and 87 in revised manuscript.

Point 18: Would be better to mention those here rather in Table 1 (line86: “other six strains were isolated from food samples that have been reported in previous study”)

Response 18: Thank you very much for your suggestion. It has been modified according to your opinion. P2 lines 90 and 91 in revised manuscript.

Point 19: Meaning??? (line 89: “ A typical colony”)

Response 19: “A typical colony” has been replaced by “a loopful of each strain”. Many thanks. P2 line 94 in revised manuscript.

Point 20: How long did it take to dry under this conditions (line 97: “The clean leaves were dried under 25℃ in the”)

Response 20: Thanks for your comment. We apologize for our unclear expression. We didn't describe it clearly before. It has been corrected. P2 line 101 in revised manuscript.

Point 21: This is not clear (line 101: “ Ethanol using resolving the supernatant retained”)

Response 21: We are so sorry for our unclear expression. It has been corrected. P3 lines 105 and 106 in revised manuscript.

Point 22: What does this mean (line 117: “tested bacteria was dripping onto the TSA plate.”)

Response 22: We are very sorry for confusing you.“tested bacteria was dripping onto the TSA plate.”has been replaced by “tested S. aureus strain (about 106 CFU/ml) was transferred into the culture medium”. P3 lines 121 in revised manuscript.

Point 23: You need to check on this amount whether it is real or a typo (line 123: “Then 125 mL”)

Response 23: Thanks very much for you reminder. It was a typo in the previous manuscript. “mL” has been replaced by “uL”.It has been corrected. P3 line 128 in revised manuscript.

Point 24: There is conflicting information about the amount of ATCE used. Is it 80 or 320mg/mL?? (line 213: “320mg/mL”)

Response 24: We apologize for confusing you. In our study, the antimicrobial activity of ATCE against S. aureus was rough dected using the diameter of inhibition zone firstly, so the amount was 320 mg/mL. Then the antimicrobial activity was further  accurately measure using minimum inhibitory concentration, so the amount was 80 mg/mL. Therefore, there is no conflicting.

Point 25: Spelling (line 230: “compered”)

Response 25: Many thanks. It has been corrected. P 6 line 245 in revised manuscript.

Point 26: Results and Discussions are not well articulated. Only listing of results and some references but no clear explaination of the effect of ATCE on microbial growth and how it can be applied in food preservation.

Response 26: Thank you very much for your precious comments on our manuscript. and agree with you on the revision of the manuscript. We have carefully revised the manuscript.

Reviewer 2 Report

The authors evaluate the antimicrobial activity and action approach of the Amaranthus tricolor crude extract against Staphylococcus aureus and the potential application in cooked meat. The manuscript is well written and the experimental plan is satisfactory.

In my opinion, the authors should better discuss the data regarding the effects of ATCE on color change and pH change.

Author Response

Response to Reviewer 2 Comments

Comments and Suggestions for Authors:

The authors evaluate the antimicrobial activity and action approach of the Amaranthus tricolor crude extract against Staphylococcus aureus and the potential application in cooked meat. The manuscript is well written and the experimental plan is satisfactory.

Point 1: In my opinion, the authors should better discuss the data regarding the effects of ATCE on color change and pH change.

Response 1: Thank you very much for your approval and precious commentsof our manuscript. We agree with you on the revision of the manuscript. Based on your precious comments, we have better discussed the data regarding the effects of ATCE on color change and pH change in the manuscript.

Reviewer 3 Report

Review of the article: " Antimicrobial Activity and Action Approach of the Amaranthus tricolor Crude Extract Against Staphylococcus aureus and Potential Application in Cooked Meat

Manuscript ID - foods-733227

I found the manuscript interesting and well prepared. Only minor changes are necessary before final acceptance of the manuscript.

Detailed comments:

Abstract - well presented, no critical remarks. However, it should be noticed that the MIC value is quite high – it is 8% (w/w)

Introduction: This part of the manuscript is quite well prepared. First of all, the authors justified the objectives of their research. However, it also should be mentioned that S. aureus is one of the  most important human and animal pathogen, including hospital or more generally healthcare-associated infections. It is not only a reason of food-born illness. The problem of drug resistance of staphylococci also should be mentioned (of course these should be short information 1-2 sentences).

There is a space lacking after steroids – line 69 (should it be catechins)

Materials and Methods

All experiments were well planned and performed.

Line 99 – why 70% ethanol (v/v) was used for extraction

Lines 108 and 109 – I am surprised that 200 ul of of ATCE was added to the 8mm filter paper disk (quite a large volume)? How was ATCE prepared. The authors obtained dry powder – was it suspended in water?

Lines 117-18 – these two sentences “Two microliters of tested bacteria was dripping onto the TSA plate. Then the tested bacteria 118 solution of S. aureus was dried and incubated at 37°C for 24 h.”

are not clear for me. I would be grateful for additional explanation (e.g. haw was it possible to to drip only 2 ul of bacteria on TSA plate). Generally I think that broth microdilution method would be better, but of course I accept the method proposed by the authors.

Line 137 - relative fluorescence units (RFUs) should be defined

Line 144 NS is not defined

As I mentioned above the MIC values is quite high. Thus,in my opinion it is important advantage of the manusript that the authors investigated not only antimicrobial effect. They also investigated if using ATCE importantly affect other properties of the product (color, pH which are importnat for consumers).

Results

Generally the results are interesting and well presented.

Line 212 – DIZs is not defined (of course, I know what it is but formally it should be explained in the text)

Figure 1 – the legend is too small – please use bigger size of the letters and symbols

Line 225 – should be S. aureus and italic (the same comment for other figures)

The results presented in figures 4, 5 and 6 suggest bactericidal (not bacteriostatic) activity - at least for concentration 2xMIC. Do you agree with my comment, if yes it should be highlighted (e.g in conclusions).

Discussion and Conclusions – well prepared. I have only noticed some typing errors in the first part of discussion: the names of Amarantus … are not written with italic, line 317 some spaces should be omitted (in bracket)  

Final opinion – minor revision

Author Response

Response to Reviewer 3 Comments

Comments and Suggestions for Authors:

Review of the article: " Antimicrobial Activity and Action Approach of the Amaranthus tricolor Crude Extract Against Staphylococcus aureus and Potential Application in Cooked Meat"

Manuscript ID - foods-733227

I found the manuscript interesting and well prepared. Only minor changes are necessary before final acceptance of the manuscript.

Detailed comments:

Point 1: Abstract - well presented, no critical remarks. However, it should be noticed that the MIC value is quite high – it is 8% (w/w)

Response 1: Thanks very much for your approval of our manuscript. We agree with you. Because ATCE is a crude extract, but its antimicrobial activity against S. aureus was worthy of recognition. Other research repotted that the MIC exhibited by Amaranthus hybridus extracts against E. coli was 453 mg/ml whereas that of Amaranthus caudatus against S. aureus was 155.6mg/ml.

Maiyo, Z.C.; Ngure, R.M.; Matasyoh, J.C.; Chepkorir, R. Phytochemical constituents and antimicrobial activity of leaf extracts of three Amaranthus plant species. African J. Biotechnol. 2010, 9, 3178–3182.

Point 2: Introduction: This part of the manuscript is quite well prepared. First of all, the authors justified the objectives of their research. However, it also should be mentioned that S. aureus is one of the  most important human and animal pathogen, including hospital or more generally healthcare-associated infections. It is not only a reason of food-born illness. The problem of drug resistance of staphylococci also should be mentioned (of course these should be short information 1-2 sentences).

Response 2: Thank you very much for your suggestion. The short information has been added to the revise manuscrupt. P2 lines 46 and 47 in revised manuscript.

Point 3: There is a space lacking after steroids – line 69 (should it be catechins)

Response 3: Thank you very much. A space has been added. P2 line 74 in revised manuscript.

Materials and Methods

All experiments were well planned and performed.

Point 4: Line 99 – why 70% ethanol (v/v) was used for extraction

Response 4: Thank you very much for your query. In the beginning, preliminary experiment was prepared to determine the appropriate extracting solution. Different concentrations of ethanol (0, 30, 50, 70, and 90%) was used to extract the plant. We finally confirm that the ATCE extracted with 70% ethanol showed better antimicrobial activity.

Point 5: Lines 108 and 109 – I am surprised that 200 ul of of ATCE was added to the 8mm filter paper disk (quite a large volume)? How was ATCE prepared. The authors obtained dry powder – was it suspended in water?

Response 5:  We are so sorry. The dry powder of ATCE was first prepared to solution than added to the 8mm filter paper disk that was cylindrical ring and was enough for 200 uL of liquid.

Point 6: Lines 117-18 – these two sentences “Two microliters of tested bacteria was dripping onto the TSA plate. Then the tested bacteria 118 solution of S. aureus was dried and incubated at 37°C for 24 h.” are not clear for me. I would be grateful for additional explanation (e.g. haw was it possible to to drip only 2 ul of bacteria on TSA plate). Generally I think that broth microdilution method would be better, but of course I accept the method proposed by the authors.

Response 6: We are really sorry that our previous expression confused you. The sentence has been rewritten as “After hardening, two microliters of tested S. aureus strain (about 106 CFU/ml) was transferred into the culture medium and waitting to dry. Then the 24-well plates were incubated at 37°C for 24 h.” The meaning was “Two microliter of  bacterium suspension was transferred by using pipetting gun, then gently touch the culture when the bacterium suspension was blowed.” P3 lines 120-122 in revised manuscript.

Point 7: Line 137 - relative fluorescence units (RFUs) should be defined

Response 7: Thanks for your reminder. It has been added. P4 lines 142 and 143 in revised manuscript.

Point 8: Line 144 NS is not defined

Response 8: We are so sorry about that the abbreviations was not defined in the first appear time. “NS” has been replaced by “sterile normal saline”. P4 line 149 in revised manuscript.

Point 9: As I mentioned above the MIC values is quite high. Thus, in my opinion it is important advantage of the manusript that the authors investigated not only antimicrobial effect. They also investigated if using ATCE importantly affect other properties of the product (color, pH which are importnat for consumers).

Response 9: We thanks very much for your comment to the improvements to the manuscript.

Results

Generally the results are interesting and well presented.

Point 10: Line 212 – DIZs is not defined (of course, I know what it is but formally it should be explained in the text)

Response 10: Thank you very much. The defined of DIZ has been added in revise manuscript. P3 line 115 in revised manuscript.

Point 11: Figure 1 – the legend is too small – please use bigger size of the letters and symbols

Response 11: Thank you very much for your reminder. Figure 1 has been modified according to your suggestion.

Point 12: Line 225 – should be S. aureus and italic (the same comment for other figures)

Response 12: Thank you very much. It has been corrected according to your comment.

Point 13: The results presented in figures 4, 5 and 6 suggest bactericidal (not bacteriostatic) activity - at least for concentration 2xMIC. Do you agree with my comment, if yes it should be highlighted (e.g in conclusions).

Response 13: Thank you very much for your comments. We also think that the bactericidal or bacteriostatic of ATCE is related to its concentration.The relevant content was added in the revised manuscript.

Point 14: Discussion and Conclusions – well prepared. I have only noticed some typing errors in the first part of discussion: the names of Amarantus … are not written with italic, line 317 some spaces should be omitted (in bracket) 

Response 14: Thank you very much for your positive comments about our manuscript. These errors have been corrected in revised manuscript.

Final opinion – minor revision

Response: We thank the reviewer for both your complimentary comments and contribution to the improvements to the manuscript. We have revised our manuscript according to your comments carefully.

Reviewer 4 Report

Comments on the paper foods-733227 sent for publication to Foods journal

This study has as objective the investigation of the antimicrobial activity and action approach of the Amaranthus tricolor crude extract against Staphylococcus aureus and potential application in cooked meat.

The paper is well written with minor issues concerning English. The experimental design sounds good. I have some minor comments that would improve the quality of the paper.

For the SDS-PAGE analyses, the quality of the gels is bad. The authors should improve the quality and analyse them further. The authors didn’t give enough explanation and discussion to the results of the gels.

The statistical analyses should be better described. Please, check also the table by giving the standard error of the means and the exact p-values for each comparison. This is also true for the figures and all the statistics.

I suggest to the authors to perform principal component analysis on their data to highlight the different treatments with the whole variables.

The figure 5 is not clear. I am wondering if it is relevant. The authors have to check this result very carefully.

In the manuscript there is some self-plagiarism that I ask to the authors to avoid, mainly in the methods.

For Table 3 and thus in the M&M, I ask the authors to compute hue-angle (h*) and Chroma (C*) and do the comparisons accordingly. The authors can refer for example to this study in Meat Science journal to compute those parameters relevant for the industry.

Gagaoua M., Picard B. & Monteils V. (2018) Associations among animal, carcass, muscle characteristics, and fresh meat color traits in Charolais cattle. Meat Sci 140, 145-56. https://doi.org/10.1016/j.meatsci.2018.03.004

Author Response

Response to Reviewer 4 Comments

Comments and Suggestions for Authors:

Comments on the paper foods-733227 sent for publication to Foods journal

This study has as objective the investigation of the antimicrobial activity and action approach of the Amaranthus tricolor crude extract against Staphylococcus aureus and potential application in cooked meat.

The paper is well written with minor issues concerning English. The experimental design sounds good. I have some minor comments that would improve the quality of the paper.

Point 1: For the SDS-PAGE analyses, the quality of the gels is bad. The authors should improve the quality and analyse them further. The authors didn’t give enough explanation and discussion to the results of the gels.

Response 1: We thank the reviewer for both your complimentary comments and contribution to the improvements to the manuscript. We have examine our SDS-PAGE imagins according to your comments carefully. There is no doubt about the imagins was some fuzzy. The reason would be voltage instability or decoloration halfway lead to picture blurring. However, we had tried our best to get those protein bands that were ideal result to analyze the contain protein of cells and as much as possible explanation and discussion the results

Point 2: The statistical analyses should be better described. Please, check also the table by giving the standard error of the means and the exact p-values for each comparison. This is also true for the figures and all the statistics.

Response 2: We thanks very much for your comments to the improvements to the manuscript. We have revised statistical analyses and check the table according to your comments carefully.

Point 3: I suggest to the authors to perform principal component analysis on their data to highlight the different treatments with the whole variables.

Response 3: Thank you very much for the suggestions. After completing more research contents in the future, we will perform principal component analysis. Because so far only two factors (concentration and treatment time) were research to discussed the action approach in our study, which were unable to perform principal component analysis.

Point 4: The figure 5 is not clear. I am wondering if it is relevant. The authors have to check this result very carefully.

Response 4: Thanks very much for your reminder. We also think the Figure 5 was unclear, but it can ba acceptable due to the bands of DNA became faint as the concentration and treatment time increase that well showed the effect of ATCE on the DNA of S. aureus. Of course, we will try our best to improve the sharpness of the pictures in the future research.

Point 5: In the manuscript there is some self-plagiarism that I ask to the authors to avoid, mainly in the methods.

Response 5: Thank you very much for your kindly advice of our manuscript. We have revised our manuscript carefully.

Point 6: For Table 3 and thus in the M&M, I ask the authors to compute hue-angle (h*) and Chroma (C*) and do the comparisons accordingly. The authors can refer for example to this study in Meat Science journal to compute those parameters relevant for the industry.

Gagaoua M., Picard B. & Monteils V. (2018) Associations among animal, carcass, muscle characteristics, and fresh meat color traits in Charolais cattle. Meat Sci 140, 145-56. https://doi.org/10.1016/j.meatsci.2018.03.004

Response 6: Thank you very much for your proposal. We have computed Chroma (C*) and hue-angle (h*) and do some explanation accordingly in the manuscript according to the references you provided.

Round 2

Reviewer 1 Report

After the authors has addressed the issues raised in the manuscript, it can now be published.